# Computer-Aided Lipase Engineering for Improving Their Stability and Activity in the Food Industry: State of the Art

**DOI:** 10.3390/molecules28155848

**Published:** 2023-08-03

**Authors:** Wenjun Cheng, Binbin Nian

**Affiliations:** State Key Laboratory of Materials-Oriented Chemical Engineering, School of Pharmaceutical Sciences, Nanjing Tech University, Nanjing 210009, China; cheng_wenjun@njtech.edu.cn

**Keywords:** lipases, foods, protein engineering, thermostability, solvent resistance

## Abstract

As some of the most widely used biocatalysts, lipases have exhibited extreme advantages in many processes, such as esterification, amidation, and transesterification reactions, which causes them to be widely used in food industrial production. However, natural lipases have drawbacks in terms of organic solvent resistance, thermostability, selectivity, etc., which limits some of their applications in the field of foods. In this systematic review, the application of lipases in various food processes was summarized. Moreover, the general structure of lipases is discussed in-depth, and the engineering strategies that can be used in lipase engineering are also summarized. The protocols of some classical methods are compared and discussed, which can provide some information about how to choose methods of lipase engineering. Thermostability engineering and solvent tolerance engineering are highlighted in this review, and the basic principles for improving thermostability and solvent tolerance are summarized. In the future, comput er-aided technology should be more emphasized in the investigation of the mechanisms of reactions catalyzed by lipases and guide the engineering of lipases. The engineering of lipase tunnels to improve the diffusion of substrates is also a promising prospect for further enhanced lipase activity and selectivity.

## 1. Introduction

With the depletion of nonrenewable resources and the increase in environmental pollution, organic reactions, which consume a lot of organic solvent s, are considered to be environmentally unfriendly [1,2]. Moreover, traditional chemical esterification often uses toxic and corrosive catalysts, such as sulfuric acid/sodium hydroxide, as well as using elevated reaction temperatures, which can also result in generating random non- selective reactions. As a practical and sustainable alternative to traditional organic reactions, enzymatic reactions exhibit extreme advantages in efficiency, stereoselectivity, and environmental friendliness [3,4,5]. Lipases, or triacylglycerol acyl hydrolases (EC 3.1.1.3), are a class of enzymes with versatile catalytic abilities in hydrolysis, alcoholysis, esterification, transesterification, and reverse synthesis reactions of triacylglycerides and some other water-insoluble esters [6,7] (Figure 1). Lipase reactions are significantly affected by their environment, including a tendency to promote ester hydrolysis at the oil–water interface, while promoting synthesis and ester exchange in the organic phase. Most recently, lipases have also been found to form C–C bonds [8,9] and degrade plastic [10] efficiently. To date, a large number of microbial lipases have been isolated and purified, and their property studies have suggested that microbial lipases have a wider range of pH and temperature actions than plant and animal lipases, high stability and activity, and specificity for substrates [11,12].

Recently, lipases have also emerged as a biocatalyst and contributor in personal care (amino acid-based surfactants), detergents, and oil production (docosahexaenoic acid (DHA) triglycerides, 1,2-Dilinoleoyl-3-Palmitoyl-rac-glycerol (OPO)), dairy products, and health fields [13,14]. Moreover, as an efficient biocatalyst, lipase has attracted more and more industrial and academic interest in food manufacturing processes, due to its robustness and efficiency in versatile reactions [15,16]. With the deepening of the concept of healthy eating, the demand for more nutritious, healthy, and tasty foods is gaining popularity, and the opportunities and challenges of food processing also are becoming more difficult. That is to say, organic solvents, toxic by-products, and bad flavor substances should be avoided as much as possible during food processing [17]. Therefore, enzymes are more and more widely used in food processing [18]. For instance, lipases have been used in meat processing [19,20], milk processing [21,22], food additive manufacturing [23,24,25], and the modification of lipids [26,27,28] and starch [29,30].

Protein engineering or, more particularly, the directed evolution and rational design of enzymes ha ve gradually received special attention due to their powerful capabilities in conferring higher activity, stability, and broader substrate selectivity to enzymes [31]. For instance, Keqin Chen and Frances H. Arnold (the latter was awarded the 2018 Nobel Prize for Chemistry for her work on directed evolution of enzymes) demonstrated that the directed evolution of protease can significantly improve its solvent resistance, which marked the first publication in the field of directed evolution [32]. More recently, to engineer enzymes for special reactions that are not accessible in nature, they expanded the concept to mechanism-guided directed evolution, and the formation of a carbon–silicon bond was achieved via the directed evolution of cytochrome c from Rhodothermus marinus [33]. Moreover, to make the screening of enzymes more efficient and low-cost, various computer-aided methods have been used in enzyme engineering, known as “rational design” [34] and “De Novo Protein Design” [35].

Over the past decades, numerous studies of enzymes used in food production have been reported, and, as a powerful tool to improve the activity, stability, and selectivity of lipases, enzyme- engineering methods have also been applied by more and more researchers in the food field [36,37]. Researchers have published several studies and reviews on the engineering of lipases [38,39]. For instance, Wu and co-workers developed a series of artificial cysteine-lipases, which showed 40-fold higher catalytic efficiency than wild-type lipase, which suggests that the active site of lipases can also be changed via directed evolution [36]. The structure and key residues of lipases that are crucial to their activity and stability have also been investigated in-depth [40,41]. Therefore, the engineering of lipases could be an efficient way to improve their performance in foods and some other fields.

However, the traditional designing of enzymes is often based on error-prone PCR (epPCR), which may involve the screening of tens or even hundreds of thousands of mutants. In this context, the utilization of computer- aided methods for the rational design of enzymes is of utmost importance. In addition, there have been few strategies and reviews published regarding the engineering of lipases based on the processing conditions and special requests in the food industry. Thus, in this systematic review, we focused on the development of computer-aided strategies in the engineering of lipases and an investigation of the structure’s relationship and key sites for regulating the enzyme’s performance during food processing.

## 2. Lipases as a Powerful Catalyst in the Food Field

As shown in Figure 2, although enzymes have been widely used in various foods, lipases are only a small part of this, which indicates that more effort should be made to promote lipase utilization in foods. In addition, as summarized in Figure 3, lipases can be used in processing many products in food industrial production. Firstly, lipases can significantly affect the flavor of meats. For instance, Dutta and co-workers suggested that immobilized lipase from Lactobacillus plantarum could be used in the degradation of meat and the synthesis of flavor esters [42]. Chen et al. added exogenous lipase during the wet-curing and dry-ripening of silver carp and found that exogenous lipase can significantly increase the free fatty acid content and the level of conjugated dienes, peroxides, and thiobarbituric acid-reactive substances, thus enhancing the flavor of the food [43]. Moreover, Shen’s group showed that the tumbling of meat can disrupt the cell structure and improve the release of lipase, leading to an increase in lipolysis in meat [44]. Chen et al. suggested that 4% NaCl can maintain lipase activity better, and thus can provide lipid sources for flavor precursors and flavor substances such as aldehydes, ketones, esters, etc., so as to form the unique flavor of Chinese sausage and improve its sensory quality [45].

Moreover, lipases can also play a key role in the synthesis of various food additives, such as sugar esters, sterol esters, vitamin esters, and others. As a kind of non-ionic surfactant, sugar esters have attracted more and more academic and industrial interest due to their high surface activity, emulsifying capacity, and biocompatibility [46]. Several lipases have been proven to have excellent performance in catalyzing the synthesis of sugar esters. For instance, Fu and Xuanxuan Lu suggested that the activity of lipases can be well maintained in ionic liquid mixtures consisting of 1-butyl-3-methyl-imidazolium acetic ([BMIm]Ac) and 1-butyl-3-methyl-imidazolium tetraflouroborate ([BMIm][BF_4_]), and starch ester can be successfully synthesized in this system with high efficiency and selectivity [47]. To make the synthesis of starch ester more sustainable, the solvent- free system was developed, and it was found that lipases can also perform very well in this system with a degree of substitution of 0.0346 [48].

To meet the various needs of different foods for oils, it is necessary to modify them. Lipases have been widely used in the modification of lipids. As shown in Table 1, several kinds of lipases can be used in the modification of lipids, especially Novozyme 435. Moreover, various kinds of reactions, such as alcoholysis, esterification, interesterification, and hydrolysis, can also be found. For instance, the hydrolysis of rapeseed oil was conducted by Delamplea and coworkers for the production of mono- and diglycerides of fatty acids; they found that a conversion of 58% can be obtained via catalyzed non-GMO lipase (supplied by Takabio) [49]. Moreover, the hydrolysis of vegetable oil via mycelium-bound lipase from filamentous fungus can also obtain a yield of polyunsaturated fatty acid of about 96.06% [50]. Lipases can also catalyze various esterification reactions in the food system. For example, the Lan group suggested that a yield of 82–95% of sugar fatty acid esters can be produced via the catalysis of Novozyme 435 in a THF/pyridine (4:1 *v/v*) solvent mixture [51]. Coincidentally, d-xylose laurate esters and l-arabinose laurate esters can be obtained via the esterification of sugars and vinyl laurate in 2-methyl-2-butanol with the catalysis of Novozyme 435 [52]. The lipase- catalyzed acylation reaction has also been widely investigated recently, and it has been found that CALB has a higher efficiency than d-Aminoacylase from *Escherichia coli*, and lipase from porcine pancreas, et al. in the acylation of fatty acids and amino acids [1,53,54].

With the deepening of investigations, researchers have also found that the catalytic efficiency of lipases, especially in the esterification reaction and acylation reaction, is significantly affected by the solvent [1,55,56,57]. Most recently, some novel green solvents were found to have great advantages in terms of the dissolution substrate and low-cost preparation, and are easy to separate and purify in enzymatic reactions. These exciting studies will undoubtedly further expand the application of lipases in the food field.

However, as foods comprise extremely complex materials, it is hard to modify them via natural enzymes at will. For instance, although sugar esters have been proven to be a series of emulsifiers with high emulsification and biosafety, their synthesis, or, more especially, their selective synthesis, is still a challenge. Different degrees of substitution (DS) of starch esters can be used in different fields. For example, high (1.5 to 3), medium (0.2 to 1.5), and low (0.01 to 0.2) DS starch esters can be used as thickeners or stabilizers in the food industry, and the development of thermoplastic and biodegradable materials, respectively [58]. SN-2 palmitate is considered to be a very important component in human milk and can be absorbed directly by the body [59]. Therefore, sn-2 monoacylglycerols are often added to human milk to modify the structure of the triacylglycerols of infant formula milk, which can reduce calcium palmitate and increase the absorption of fatty acids [60]. However, few enzymes can be used in the synthesis of sn-2 monoacylglycerols, and, of these, Lipozyme TL IM [61] and Lipozyme RM IM [62,63] are the most popular and efficient. Lipozyme 435 [63] and NS40086 [64] from Novozymes (Denmark), and lipase DF-15 from Amano (Japan) can also be used in the synthesis of sn-2 monoacylglycerols; however, their efficiency is extremely low due to the large spatial bit resistance [61]. In this context, the development of new enzymes and the engineering of natural enzymes are of utmost importance in the food and health fields.

**Table 1 molecules-28-05848-t001:** The applications of lipases in the modification of lipids.

Substrates		Enzymes	Solvents	Time (h)	Products	Yield	References
Acyl Aceptor	Acyl Donor						
Rapeseed oil	/	Non-GMO lipase came from Takabio	Water	24	Mono- and diglycerides of fatty acids	56%	[49]
Vegetable oils	/	Mycelium-Bound lipase from *filamentous fungus*	Sodium phosphate buffer	9	polyunsaturated fatty acids	96.06%	[50]
Sesame oil fatty acids	/	Lipozyme RM IM	Water	2	sn-2 PA	67.70%	[65]
Phenolic acid ethyl esters	Glycerin	Novozym 435	Glycerin	4	monoferuloyl glycerol	91.60%	[66]
Triolein	Olive oil	*Yarrowia lipolytica* lipase	Solvent-free	0.25	Structured lipids	33%	[67]
Solid β-sitostanol	Lauric or oleic acid	*Ophiostoma piceae* lipase	Isooctane/water biphasic systems	3	β-sitostanol esters	90%	[68]
Xylobiose	Vinyl laurate	Lipase N435	2-Methyl-2-butanol	72	4′-*O*-laurylxylobiose	86%	[69]
Hexyl alcohol	Octanoic acid	*Candida rugosa* lipase	Isooctane	0.67	hexyl octoate	50%	[70]
Glycerol	Conjugated linoleic acid	Lipases B from *Candida antarctica*	Acetone	3	partial glycerides of conjugated linoleic acids	54%	[71]
Cyanidin-3-*O*-galactoside	Saturated fatty acids	Novozyme 435	Tert-butanol	72	cyanidin-3-*O*-(6″-dodecanoyl) galactoside	73%	[72]
d-glucose	Vinyl hexanoate, vinyl octanoate	Novozyme 435	THF/pyridine (4:1 *v/v*)	48	Sugar fatty acid esters	82–95%	[51]
d-xylose, L-arabinose	Vinyl laurate	Novozyme 435	2-Methyl-2-butanol	0.17	d-xylose laurate esters, l-arabinose laurate esters	53%, 48.6%	[52]
*A. tequilana* fructans	Vinyl laurate	Novozyme 435	Hexane	96	Carbohydrate fatty acid esters	80%	[73]
Lysine	lauric acid	Lipases B from *Candida antarctica*	Deep eutectic solvents	120	Lipoamino acids	59.64%	[1]

In other words, although lipases have shown so many fascinating advantages, their low efficiency, low selectivity, and low solvent adaptability in some reactions also limit further applications in the food field [36,74,75]. Therefore, the engineering of lipases to improve their efficiency, solvent resistance, thermal stability, and substrate selectivity has attracted a lot of academic and industrial interest in the past twenty years [76,77]. In the last two decades, numerous attempts have been put forward for the design and evolution of lipases, which have promoted the industrial application of lipases in food fields.

## 3. The Applications of Computer-Aided Methods in Designing and Engineering Lipases

The authors will discuss the results and how they can be interpreted from the perspective of previous studies and the working hypotheses. The findings and their implications should be discussed in the broadest context possible. Future research directions may also be highlighted.

### 3.1. General Structure of Lipases

Up to now, about 4300 lipase structures (accounting for 2% of all the proteins in this data bank) have been deposited in the RCSB protein data bank (https://www.rcsb.org/ (accessed on 16 July 2023)). As shown in Figure 4, the resolution of structures has increased exponentially since the 1980s, indicating that this field is receiving increasing attention. Moreover, we can also see that the lipase from Mus musculus was one of the most studied lipases in recent decades. The RCSB protein data bank can not only provide information about the source and date of lipase discoveries but also can help us to understand their structures. Although the amino sequence and substrate selectivity of lipases differ from one to another, their configurations are extremely similar [12]. The crystal data from XRD, cryogenic electron microscopy, and NMR suggested that lipases belong to the family of α/β hydrolase. The various amounts and sites of β sheets and α helixes have endowed them with a wide scope of substrate selectivity (Figure 5a) and their “barrel structure” (Figure 5b) [78].

In addition, most lipases have a lid in their structure which can protect the active site from solvents, heating, and some other unfavorable factors (Figure 5c) [79]. There are about two kinds of lids in lipases: one of them consists of one or two amphipathic α helixes, and the other one consists of a loop structure [74,80]. The lid regulates “closed” and “open”, allowing the entry of substrates to the active site (Ser-His-Asp or Ser-His-Glu; the nucleophilic Ser is located in the highly conserved G-X-S-X-G sequence), where the lipases are exposed to the oil–water interface. For instance, the coarse-grained molecular dynamic study of M37 lipase (PDBID: 2ORY) suggests that interactions with triglyceride surfaces can promote the motion of the lid region (residue 235–283) of lipases and provide a pathway for substrates [81]. The introduction of glycine around the active site of *Geobacillus thermocatenulatus* lipase can significantly enhance the specific activity compared to wild type at temperatures between 283 K and 363 K with p-nitrophenol butyrate [82]. 

B iocatalytic reactions normally involve several steps for substrates transporting and binding to the active sites of lipases and the releasing of products. However, the active sites of most lipases are usually located in non-exposed areas in the protein structure, and the substrates can only be transported via tunnels. Therefore, the substrate tunnel can also play a key role in the activity of lipases, which can offer a unique microenvironment for biological functions, such as ligand binding or enzymatic catalysis [83]. Moreover, the tunnel can also enhance the substrate scope of enzymes. For instance, although the XRD structural studies of *Mycobacterium tuberculosis* lipases suggested that the active site accommodates up to thirteen carbon atoms, the experimental results suggested that the long chain fatty acids of 16–18 carbon atoms have the best affinity to this enzyme, which can be attributed to the high flexibility of the loop structure located at the entry of the substrate tunnel [84]. Besides transporting and controlling the substrates, ligands, or products, the tunnel structure can also modulate the activity and selectivity of enzymes by avoiding the formation of unfavorable intermediates [85], and, in most cases, there are several tunnels in one lipase that can cooperate in enzymatic reactions [86,87,88].

### 3.2. Computer-Aided Methods for the Prediction of Lipases Structures

Although some lipase structures have been decrypted, most of them are still mysteries, due to great difficulty in obtaining protein crystals and the data analysis of XRD, HNMR, and cryogenic electron microscopy. To date, several strategies have been developed for the prediction of enzyme structures based on computer-aided technologies. Among them, AlphaFold2 is the most famous one, which gained significant attention and acclaim after its successful performance in the Critical Assessment of Structure Prediction (CASP) Competition in 2020, where it outperformed other methods by a substantial margin. AlphaFold2 utilizes deep learning and artificial intelligence techniques to predict the three-dimensional structures of proteins. Alphafold2’s innovative approach combines two key components: a novel representation of protein structures as distances and orientations between pairs of amino acids, and an attention-based neural network architecture that effectively models the complex dependencies between these pairs. The training of Alphafold2 involved learning from a vast amount of protein structure data, making it capable of predicting highly accurate structures for a wide range of proteins, including those with limited experimental data. To date, Alphafold2 has been widely used in the prediction of lipase structures. For instance, Ofiţeru et al. suggested that AlphaFold2 could be used for the identification and classification of millions of protein sequences from metagenomics sequencing, thus obtaining 78 putative cold-adapted bacterial lipase sequences [89]. Moreover, Oberer and co-workers indicated that three-dimensional models of adipose triglyceride lipases demonstrated that the mapping of important amino acids can strongly corroborate the significance of these residues [90].

Besides Alphafold2, the I-TASSER (Iterative Threading ASSEmbly Refinement) has also emerged as a powerful tool for protein structure prediction by integrating multiple computational techniques and diverse experimental information [91]. Developed by the Zhang lab at the University of Michigan, I-TASSER employs a hierarchical approach to predict protein structures, encompassing several stages of assembly, refinement, and optimization. At its core, I-TASSER utilizes threading algorithms to generate initial structural models based on evolutionary information and threading templates. These models are then subjected to iterative refinement using molecular dynamic simulations and atomic-level energy minimization. Furthermore, I-TASSER incorporates additional information such as predicted contacts, secondary structure, and solvent accessibility to improve the accuracy of the final models.

In addition to the above- mentioned, so- called ab initio modeling, homology modeling has received more and more attention, due to its extremely low computational load and high accuracy. SWISS-MODEL represents a pioneering approach in the field of protein structure modeling, focusing on the creation of high-quality homology models [92]. Developed by the Swiss Institute of Bioinformatics, SWISS-MODEL excels at predicting structures for proteins that share sequence similarity with proteins of known structures, leveraging the principle that homologous proteins are likely to have similar structures and functions. SWISS-MODEL operates by searching a vast database of experimentally determined protein structures and selecting the most suitable templates for modeling a target protein. These templates are then aligned with the target protein sequence, and the resulting alignment is used to generate a three-dimensional model by transferring the known structural information. SWISS-MODEL provides a comprehensive suite of tools to assess model quality, enabling researchers to make informed decisions based on reliability and accuracy.

### 3.3. Protein Engineering Strategies

In recent decades, lipases have attracted widespread academic and industrial interest due to their efficiency, robustness, solvent tolerance, selectivity, and thermostability [79]. However, as natural biocatalysts, lipases suffer some drawbacks, including poor stereo- and regioselectivity, a narrow substrate spectrum, low catalytic efficiency, poor thermostability and product inhibition, etc. [76,93,94]; these have undoubtedly limited their further utilization in food fields. The evolution of enzymes in nature always takes thousands of years, it is a long and unpromising process. In recent decades, the new generation of protein engineering technologies, such as saturation mutations, error-prone PCR (epPCR), and DNA pooling executing much higher mutation and recombination rates to screen for the desired biological function, has developed. Figure 6 summarizes t he simplified protocol of protein engineering. Protein engineering generally includes three strategies: random mutagenesis, semi-rational design, and rational design [95]. The general idea is to repeat rounds of mutation, expression, and screening of the target gene so that the evolution that takes thousands of years in nature can be completed in a short period of time, and, finally, a protein with improved performance or a new function can be obtained [96].

The random mutagenesis (non-rational design) strategy can generate numerous beneficial mutations via its high-throughput screening methods and the fact that no in-depth knowledge of the enzyme sequence and structure is required [97]. The biggest obstacle to non-rational design is the construction of high-throughput screening methods and the screening of tens of thousands of mutant libraries, which consume a lot of human, material, and financial resources. To make these methods more efficient, Ulrich Schwaneberg developed Sequence saturation mutagenesis (SeSaM), which can efficiently disrupt the preference of DNA polymerase and improve mutation efficiency [98]. The rational design method differs from non-rational design in that it requires a deep understanding of the structure of the enzymes and further identifies which residues have a greater impact on the properties of the enzymes. However, although the development of bioinformatics [99], molecular dynamic study [100], machine learning [101], and some other computer- aided technologies [101,102,103,104,105] have significantly improved the efficiency of rational design, it still retains a lower success rate compared to non-rational design, due to its high dependence on computer resources and the lower stability of newly designed enzymes. Semi-rational design, which is based on the enzyme structure and site- saturation mutagenesis for constructing the small and refined mutation library, is one of the most popular strategies in enzyme engineering [106,107,108]. Compared to rational design and non-rational design, semi-rational design exhibits great advantages in terms of the smaller mutation library and high-benefit mutations and has attracted more and more research interest [109,110].

### 3.4. Computer-Aided Methods for the Thermostability Engineering of Lipases

Although most industrial applications of lipases need to be conducted at high temperatures (normally above 60 °C), the optimum temperature of most lipases is below 50 °C. For instance, the optimum temperatures of *Penicillium cyclopium* PG37 [111], *Penicillium expansum* lipase [112], and *Candida rugosa* lipase [113] are 25 °C, 34 °C, and 40 °C, respectively. This limits their applications in food and some other fields. The utilization of protein engineering technology to improve the thermostability of lipases in these fields has been an ongoing objective [114]. In general, there are several practical strategies that can be used for improving lipase thermostability (Figure 7): (1) the introduction of non-covalent interactions (hydrogen bonds (hbonds), salt bridges, π-π stacking, hydrophobic interaction, etc.); (2) the addition of covalent interactions (disulfide bonds); (3) an increase in proline and/or decrease in glycine; (4) the reinforcement of subunit–subunit interactions [115], and (5) ancestral sequence reconstruction. The formation of non-covalent interactions is one of the best ways to improve the stability of enzymes. Mitra suggested that the introduction of glutamic acid and aspartic acid can significantly enhance the hbond interactions and rigidity of *Bacillus subtilis* lipase, and thus improve the optimum temperature of activity from 35 °C to 65 °C, and the MD simulations indicated that thermostable mutants can maintain their structures via activating some essential modes in the catalytically relevant loop regions [116]. In addition, the mutation of G14S and V109S introduces new hbonds in the structure of *Bacillus pumilus* lipase and increases its t_1/2_ at 50 °C for 9.2 and 8.8 folds [117]. The introduction of cysteine at residues 190 and 238 will lead to the formation of a new disulfide bond in *Rhizopus oryzae* lipase, increase its rigidity, and lead to its t_1/2_ at 55 °C and 60 °C increasing about 102.5 and 20 folds [118]. This suggests that the introduction of covalent interactions may be more beneficial to the thermostability of enzymes, and some other studies also demonstrate that more disulfide bonds are beneficial to the resistance of enzymes. However, we cannot introduce too many disulfide bonds in one enzyme, due to the fact that it can lead to the incorrect folding of enzymes. Gillam et al. suggested that ancestral sequence reconstruction can generate highly stable folds and thus improve the thermostability of enzymes since ancestral proteins were frequently more thermostable than their extant descendants [119]. To date, the ancestral sequence reconstruction method has been used for improving the stability of many enzymes, such as lipases, nucleoside diphosphate kinase, 3-isopropylmalate dehydrogenase, etc. [120,121]. The main processes of ancestral sequence reconstruction mainly include: the nucleic acid/amino acid sequence collection of modern enzymes, multiple sequence alignment, phylogenetic tree construction, computer prediction of ancestral enzyme sequences, gene cloning, and enzymatic property characterization [122]. Among them, the ClustalW and MUltiple Sequence Comparison by Log-Expectation (MUSCLE) are the most widely used multiple sequence alignment programs, and the MUSCLE exhibits great advantages in speed and accuracy.

Nowadays, with rapid development in computer technology, more and more efforts have been put forward in the computer- aided design of thermostability enzymes, and some excellent programs, such as PoPMusic [124], FoldX [125], MODIP [126], and FireProt [127], have been released [127]. Among them, the FireProt web server has attracted widespread interest due to its advantages of being easy to use, with a novice-friendly interactive interface, and powerful and diverse functions; we will introduce its utilization to promote its further application in enzyme design. For instance, eight-point mutations for improving the thermostability of serine protease from *Pseudomonas aeruginosa* were predicted via Fireprot, and A29G and V336I yielded a positive impact, which increased the Kcat/Km ratio to 1.5- and 1.8- fold compared to the wild type [114]. Moreover, Chen suggested that combining FireProt, structure analysis, and MD simulations would provide more benefit in engineering thermostable enzymes, and, based on this strategy, T219L and I31F/T219L/T263L/S360A were obtained with an increase of two folds of half-lives at 50 °C compared to the wild type [128].

### 3.5. Computer-Aided Methods for Solvent Tolerance Engineering

The benefit of non-aqueous enzymatic reactions has strengthened the engineering and screening of solvent- tolerant enzymes [129]. In recent decades, numerous efforts have been dedicated to improving the solvent resistance of various enzymes. Lipases are some of the most widely-used biocatalysts in non-aqueous enzymatic reactions, due to their relatively high stability in organic solvents. Haiyang Cui et al. suggested that compared to non-polar solvents, polar organic solvents can penetrate deeper into enzymes and interact with key residues, such as binding pockets and active sites, and thus lead to a dramatic change in the structure of enzymes [77]. Solvents can regulate the stability and activity of enzymes from five aspects: (1) competing for adsorption- bound water, which is crucial for enzyme activity [130], (2) restricting conformational flexibility [54,130,131,132], (3) competitive inhibition binding of substrates [133], (4) affecting substrate solubility and mass transfer effects [134], and (5) stabilizing transition states [134]. An aqueous solution is the most suitable medium for most natural enzymes. However, the low solubility and mass transfer effect of many poly substrates limits their applications in many fields.

In this context, the engineering of solvent-tolerant enzymes is one of the most interesting directions to promote further applications of lipases in foods and other fields. As shown in Table 2, several studies on the engineering of lipase solvent tolerance are displayed and suggest that there are several strategies that can be used to improve the solvent tolerance of lipases. Many previous studies have suggested that the formation of new h-bonds is beneficial to the organic solvent tolerance of lipases. For instance, Oghino suggested that the half-life of A28S variants of Thermomyces lanuginosus lipase was increased to 6.7-folds longer than the wild type, due to the newly formed h-bonds between A28S and D274 [135]. Zhu indicated that the substitutions of charged residues will improve the solvent resistance of lipases [136]. Most recently, Haiyang Cui found that the essential water [137] and hydration shell [4] of lipase play a predominant role in the resistance of organic solvents, and the introduction of charged residues on the surface can increase the essential water molecules locally. Moreover, they argued that although salt bridges can improve enzyme stability in most cases [138,139], an unfavorable salt bridge may also lead to a strong interaction with solvents, and the introduction of oppositely charged amino acids can cause the unfavorable salt bridge to be disrupted, and thus improve the solvent tolerance [140]. Frauenkron-Machedjou revealed two important principles for improving the ionic liquid resistance of lipases: (1) Substitutions of 50–69% of the amino acid positions can improve the ionic liquid (ILs) resistance. (2) The substitutions of chemically different amino acids (such as from aromatic to polar/aliphatic/charged amino acids) are beneficial to the ILs resistance [141]. To unravel the effects of various amino acid substitutions on the stability and activity of enzymes, Zhao et al. studied several previously reported *Bacillus subtilis* lipase A (BSLA) variants with MD simulations and suggested that the introduction of positively charged amino acids close to the substrate-binding cleft can reduce the probability of ILs cations, and thus improved the ILs resistance [142]. Based on this theory, they successfully identified nine IL- resistant BSLA lipase variants, and M134R/L140S showed almost doubled solvent resistance from the wild type [143]. Moreover, El Harrar et al. argued that the long-range perturbation pathways will also significantly affect the ILs resistance of lipase [144]. To make engineering IL- resistant lipases easier and more universal, they developed a critical assessment of structure-based approaches and found that the prediction precision can reach 90% [145]. By investigating the molecular knowledge about how substitutions to charged amino acids improve enzyme activity via MD simulations, Haiyang Cui suggested that the introduction of both negatively and positively charged amino acids was beneficial for increasing essential water molecules [137]. This is different from some previous studies, which elucidated that the anions of ILs play a key role in the regulating of enzyme activity and stability. In summary, there are some principles for the engineering of IL- resistant lipases: (1) the introduction of charged residues on the surface of enzymes; (2) the introduction of positively charged amino acids close to the substrate-binding cleft; and (3) the formation of disulfide bond or hydrogen bond/salt bridges.

## 4. Conclusions and Outlook

Lipases are one of the most widely used biocatalysts and can be used in esterification, amidation, hydrolysis, and transesterification reactions. However, natural lipases have drawbacks in terms of their resistance to organic solvents, thermostability, selectivity, etc., which limit some of their applications in the food field. In recent decades, protein engineering, or, more especially, the directed evolution and rational design of enzymes, has gradually been receiving special attention, due to its powerful capabilities in conferring higher activity, stability, and broader substrate selectivity on enzymes. Recently, the protein engineering of lipases has attracted more and more academic and industrial interest and achieved some milestones, such as that organic solvents and IL- resistance have been significantly improved via rational design and directed evolution. Some engineering principles have been proposed, including several practical strategies that can be used for improving lipase thermostability: (1) the introduction of non-covalent interactions (hbonds, salt bridges, π-π stacking, hydrophobic interaction, etc.); (2) the addition of covalent interactions (disulfide bonds); (3) an increase in proline and/or decrease in glycine; (4) the reinforcement of subunit–subunit interactions. There are some principles for engineering IL- resistance in lipases: (1) the introduction of charged residues on the surface of enzymes; (2) the introduction of positively charged amino acids close to the substrate-binding cleft; and (3) the formation of disulfide bonds or hydrogen-bond/salt bridges.

Although many studies have been conducted in the field of lipase engineering, these tasks are still partially blind, and it is urgent to elucidate the mechanism by which lipases target specific reactions by computer- aided technology in the future. Moreover, how the substrates enter and exit the active site is also ambiguous, and the engineering of lipase tunnels to improve the diffusion of substrates is also a promising prospect for further enhanced lipase activity and selectivity.

## Figures and Tables

**Figure 1 molecules-28-05848-f001:**
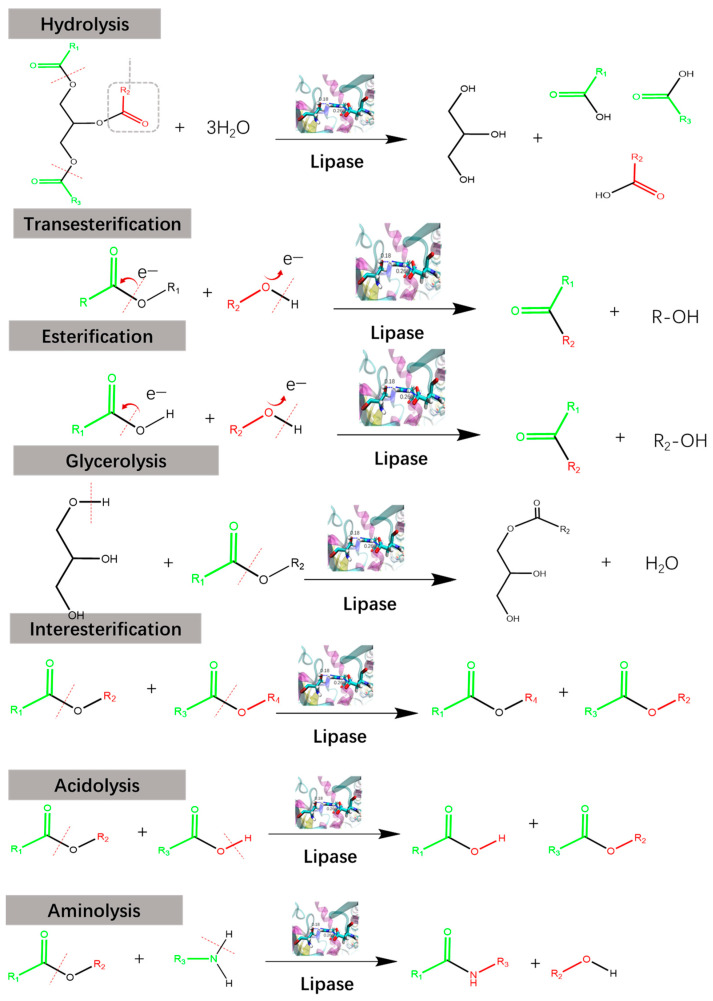
The various reactions catalyzed by lipases.

**Figure 2 molecules-28-05848-f002:**
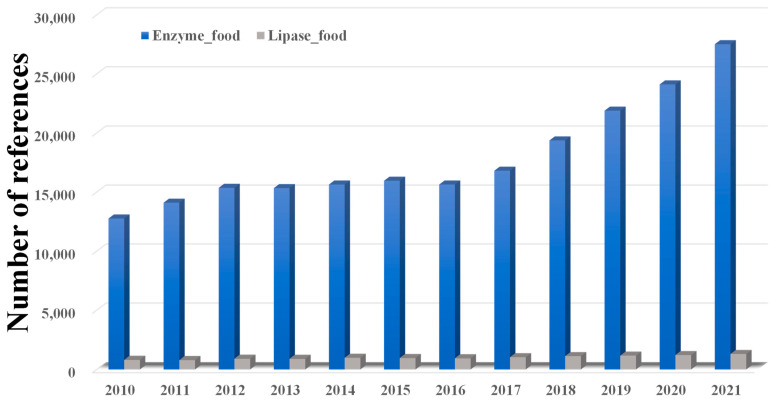
Number of publications in the fields of enzymes and foods from 2010 to 2022 (searched via the Web of Science, 22 July 2023). 319,858 publications can be searched in the database of the Web of Science for the topic of enzymes and foods, and 18,102 of those publications are related to lipases and foods.

**Figure 3 molecules-28-05848-f003:**
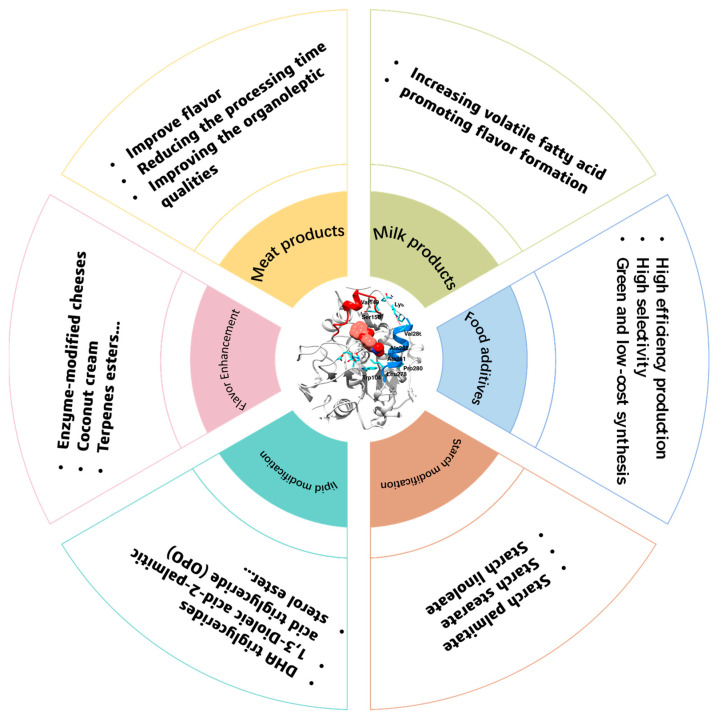
The main applications of lipases in the food industry.

**Figure 4 molecules-28-05848-f004:**
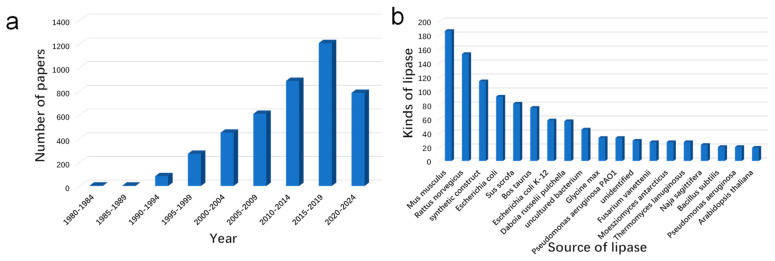
The summary of lipases in the RCSB protein data bank from 1980 to 2022, the number of papers (**a**), and the kinds of lipases (**b**) (searched by the web server of https://www.rcsb.org/ (accessed on 22 September 2022)). 4301 lipases can be found in the RCSB protein data bank by searching the keyword “lipase”, and the data of the date and microbial source were directly exported from the RCSB database.

**Figure 5 molecules-28-05848-f005:**
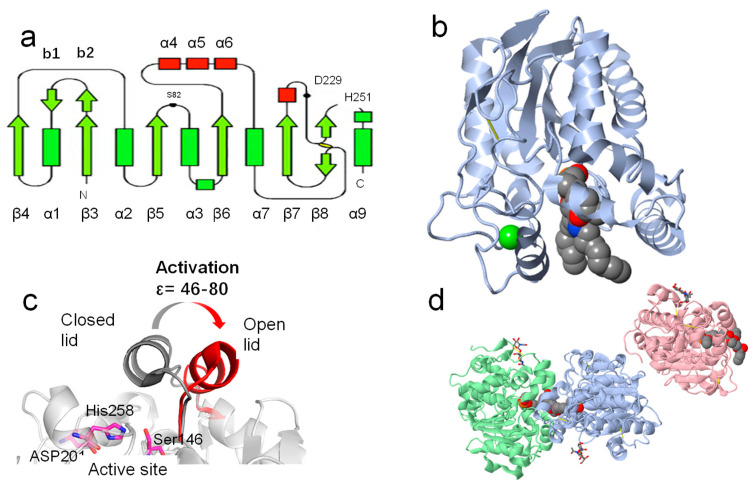
The β sheets and α helixes in lipases. Normally, there are eight β sheets in lipases (**a**). The classical “barrel structure” of *Pseudomonas aeruginosa* lipase (PDBID:1EX9) (**b**). The lid of *Thermomyces lanuginosus* lipase (active site triad: Ser146, His258, and Asp201), and the ligands were drawn in Vdw style (**c**). The no- lid structure of *Candida antarctica* lipase B (PDBID: 6TP8) and the active site residues are colored in pink, and the ligands are drawn in Vdw style (**d**), copyright 2016, American Chemical Society.

**Figure 6 molecules-28-05848-f006:**
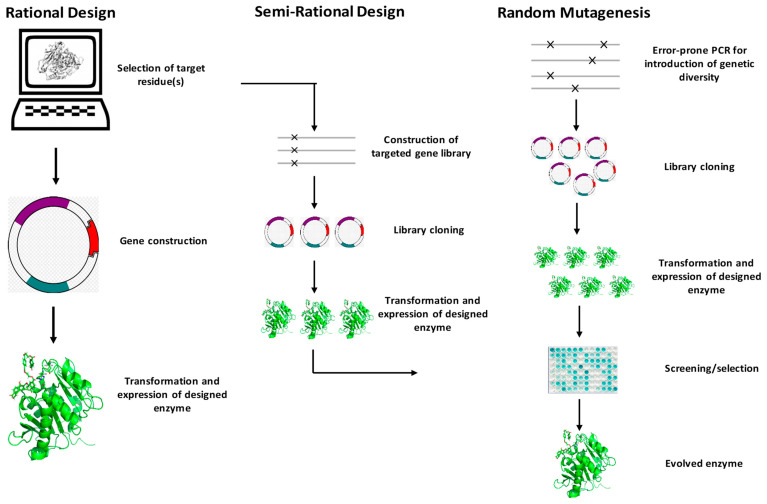
The summary of the strategies of protein engineering.

**Figure 7 molecules-28-05848-f007:**
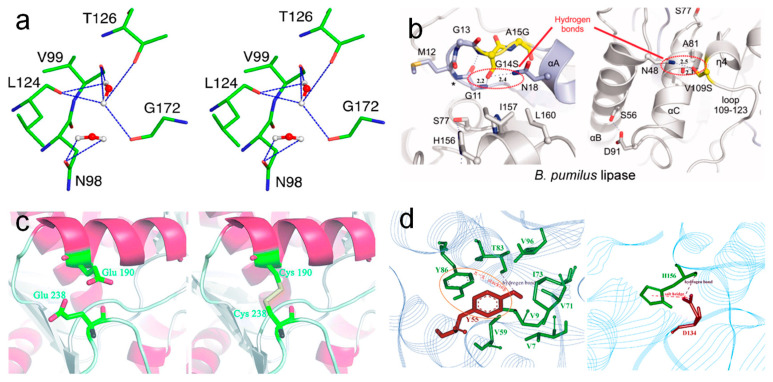
Stereo view of the snapshot from the simulation of M134E-M137P-S163P mutant of *Bacillus subtilis* lipase at 450 K and the hbonds stabilized via water. Amino acid residues and solvent molecules are shown in stick and ball-stick representations, respectively (**a**) [116], copyright 2015, American Chemical Society; the V109S point mutation of *Bacillus pumilus* lipase and the formation of hbonds (**b**) [117], copyright 2013, Elsevier; interactions around the mutated positions of wild-type *Rhizopus oryzae* lipase and the equivalent positions in the E190C/E238C variant (**c**) [118], copyright 2018, RSC; the local structures of mutated Y55 and D134 of *Bacillus subtilis* lipase A and the formation of π-stacking interactions (**d**) [123], copyright 2014, Springer-Verlag Berlin Heidelberg.

**Table 2 molecules-28-05848-t002:** Several studies of the engineering of solvent tolerance of lipases.

Enzymes	Substrate	Solvents	Substitutions/Modifictaion Region	Protein Engineering Strategy	Yields/Conclusion	References
*Bacillus subtilis* Lipase A	*p*-Nitrophenyl butyrate	Ionic liquids	Aromatic/aliphatic/polar/charged amino acids	Multiple site saturation mutagenesis	6–13% of substitutions of these sites can improve resistance	[141]
*Bacillus subtilis* Lipase A	*p*-Nitrophenyl butyrat*e*	Ionic liquids-water	M134N, N138S, L140S	Multiple site saturation mutagenesis	Variant M2 (M134R/L140S) showed almost double resistance (233% vs. 111%) of ILs co-solvent	[142]
*Bacillus subtilis* Lipase A	*/*	Ionic liquids	Perturbation pathways	Molecular dynamics simulations	Identifying these perturbation pathways and specific IL ion-residue interactions there effectively predicts focused variant libraries with improved ILs tolerance.	[144]
*Bacillus subtilis* Lipase A	*/*	Ionic liquids	/	Molecular dynamics simulations	The combination of five simple-to-compute physicochemical and evolutionary properties (P9-P12) substantially increases the precision of predicting relevant variants and positions of BsLipA for increasing aIL resistance	[145]
*Thermomyces lanuginosus* lipase	BAL-tributyrate	50% (*v/v*) methanol	A28S	Site-directed mutagenesis	Half-life was 6.7-fold longer than wide-type in 50% (*v/v*) methanol at 40 °C	[135]
*Bacillus subtilis* Lipase A	*p*-Nitrophenyl butyrate	Trifluoroethanol/1,4-dioxane/dimethyl sulfoxide	/	site saturation mutagenesis	Charged substitutions on the surface predominantly improved resistance while polar ones were preferred in buried “regions”	[136]
Bacillus subtilis Lipase A	*p*-Nitrophenyl butyrate	1,4-dioxane/ dimethyl sulfoxide	Salt bridge	Molecular dynamics simulations + site saturation mutagenesis	The variants of organic solvents can be improved 7.6-fold	[140]

## Data Availability

All data are either cited and/or included in the main text.

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
