# Peer review of "Computer-Aided Lipase Engineering for Improving Their Stability and Activity in the Food Industry: State of the Art"

_molecules, 2023, doi:10.3390/molecules28155848_

Round 1
Reviewer 1 Report
This manuscript is well written, but could be improved with the following corrections or additions. Please consider the following.
1. Please recheck the English throughout the manuscript.
2. All figures are blurred. Please change them to clear illustrations.
3. Scientific names of many organisms are not italicized. Please italicize the scientific names throughout the manuscript.
4. Please lowercase the "l" in lipase except for lipase at the beginning of sentences and the commercial name of lipase.
5. When three or more words are placed side by side, such as A, B, and C, a comma is usually placed before the "and", such as A, B, and C.
6. P3, L2. Please remove "et al.
7. P3. "Chen et al., added" should be "Chen et al. added". Please delete the comma after "et al.".
8. P5: It may be difficult to understand the meaning of "0.0346".
9. P5. "d-" and "l-" should be small capital letters.
10. P5. Vinyl laurate" should be " vinyl laurate". 2-Methyl-2-butanol" should be "2-methyl-2-butanol".
11. Table 2. "Multiple Site Saturation Mutagenesis" should be "Multiple Site Saturation Mutagenesis". The bottom of " site saturation mutagenesis" should be " Site saturation mutagenesis". Please define "ILs" first somewhere like Ionic liquids (ILs).
12. P8, L24. "et al."?
13. Fig. 4. Please label the two figures "a" and "b" and explain each in the legend. Please describe the units on the vertical axis.
14. Fig. 5b. Please explain the meaning of colored moleculs in the legend.
15. Fig. 5c. Remove the words "Thermomyce lanuginosus lipase" from the figure. This is not necessary as it is described in the legend. If possible, please describe which is Ser146, His258, or Asp201 in the figure. Please briefly explain what the figure means in the legend.
16. Fig. 5d. "d" is not shown in the legend. Please explain the meaning of the figure in the legend.
17. Fig. 7. Please explain each figure in the legend.
18. 3.2. Computer-aided methods for prediction of lipase structure.
The authors mention only the Swiss model, but it would be better to add descriptions of prediction of lipase structure using Alphafold2.
19. 3.4, Computer-aided methods for thermostability engineering of lipase.
It would be better to add descriptions of ancestral mutation methods, etc. (https://doi.org/10.1016/j.jbc.2022. 102435).
20. L169. Please first define "hbonds" somewhere like "hydrogen bonds (hbonds)".
21. L246. What is "romantic"? aromatic?
Understandable, but needs to be corrected in minor details.
Author Response
Reviewer 1:
This manuscript is well written, but could be improved with the following corrections or additions. Please consider the following.
- Please recheck the English throughout the manuscript.
Thanks so much for your kindly comments, the English has been thoroughly revised in this version.
- All figures are blurred. Please change them to clear illustrations.
Thanks so much for your kindly comments, the Figures may be compressed during the process of submission, and we have change them by high-submission ones in this version.
- Scientific names of many organisms are not italicized. Please italicize the scientific names throughout the manuscript.
Thanks so much for your kindly comments, the scientific names of all organisms involved in this paper were revised to italics in this version.
- Please lowercase the "l" in lipase except for lipase at the beginning of sentences and the commercial name of lipase.
Thanks so much for your kindly comments, all of the "l" in lipase except for lipase at the beginning of sentences and the commercial name was revised to lowercase.
- When three or more words are placed side by side, such as A, B, and C, a comma is usually placed before the "and", such as A, B, and C.
Thanks so much for your kindly comments, we have check and revised it throughout the paper in this version.
- P3, L2. Please remove "et al.
Thanks so much for your kindly comments, we have check and revised it in this version.
- "Chen et al., added" should be "Chen et al. added". Please delete the comma after "et al.".
Thanks so much for your kindly comments, we have check and revised it in this version, Moreover, some other similar errors have also been revised.
- P5: It may be difficult to understand the meaning of "0.0346".
Thanks so much for your kindly comments, “0.0346” is the value of the degree of substitution of starch.
- "d-" and "l-" should be small capital letters.
Thanks so much for your kindly comments, “l-arabinose laurate esters” and “d-Aminoacylase” were revised to “L-arabinose laurate esters” and “D-Aminoacylase” in this version.
- Vinyl laurate" should be " vinyl laurate". 2-Methyl-2-butanol" should be "2-methyl-2-butanol".
Thanks so much for your kindly comments, “Vinyl laurate” and “2-Methyl-2-butanol” were revised to “vinyl laurate” and “2-methyl-2-butanol” in this version. Moreover, some other similar errors have also been revised.
- Table 2. "Multiple Site Saturation Mutagenesis" should be "Multiple Site Saturation Mutagenesis". The bottom of " site saturation mutagenesis" should be " Site saturation mutagenesis". Please define "ILs" first somewhere like Ionic liquids (ILs).
Thanks so much for your kindly comments, Table 2 was revised according to your suggestions in this version. Moreover, some other similar errors have also been revised. The full name of ILs has also been added in P17.
- P8, L24. "et al."?
Thanks so much for your kindly comments, it has been deleted in this version.
- 4. Please label the two figures "a" and "b" and explain each in the legend. Please describe the units on the vertical axis.
Thanks so much for your kindly comments, Figure 4 was revised and the units on the vertical axis were added.
- 5b. Please explain the meaning of colored moleculs in the legend.
Thanks so much for your kindly comments, the meaning of colored moleculs in the legend was added in this version.
The β sheets and α helixes in lipases. Normally, there are eight β sheets in lipases (a). The classical “barrel structure” of Pseudomonas aeruginosa Lipase (PDBID:1EX9) (b). The lid of Thermomyces lanuginosus lipase (active site triad: Ser146, His258, and Asp201), and the ligands were drawn in Vdw style (c). The no lid structure of Candida antarctica Lipase B (PDBID: 6TP8) and the active site residues were clored in pink, and the ligands were drawn in Vdw style.
- 5c. Remove the words "Thermomyce lanuginosus lipase" from the figure. This is not necessary as it is described in the legend. If possible, please describe which is Ser146, His258, or Asp201 in the figure. Please briefly explain what the figure means in the legend.
Thanks so much for your kindly comments, Figure 5 was revised as your suggestion in this version.
- 5d. "d" is not shown in the legend. Please explain the meaning of the figure in the legend.
Thanks so much for your kindly comments, the legend of Figure 5d was added in this version.
The no lid structure of Candida antarctica Lipase B (PDBID: 6TP8) and the active site residues were clored in pink, and the ligands were drawn in Vdw style (d)
- 7. Please explain each figure in the legend.
Thanks so much for your kindly comments, the explanation of each figure in Figure 7 was added in the legend.
Stereo view of the snapshot from the simulation of M134E-M137P-S163P mutant of Bacillus subtilis Lipase at 450 K and the hbonds which stabilized via water. Amino acid residues and solvent molecules are shown in stick and ball-stick representation, respectively (a)[112], Copyright 2015 American Chemical Society; The V109S point mutation of Bacillus pumilus lipase and the formation of hbonds (b)[113], Copyright 2013 Elsevier; Interactions around the mutated positions of wide-type Rhizopus oryzae lipase and the equivalent positions in the E190C/E238C variant (c)[114], Copyright 2018 RSC; The local structures of mutated Y55 and D134 of Bacillus subtilis lipase A and the formation of π-stacking interactions (d)[137], Copyright 2014 Springer-Verlag Berlin Heidelberg.
- 2. Computer-aided methods for prediction of lipase structure.
The authors mention only the Swiss model, but it would be better to add descriptions of prediction of lipase structure using Alphafold2.
Thanks so much for your kindly comments, the descriptions of prediction of lipase structure using Alphafold2 was added in this version.
Up to date, Alphafold2 has been widely used in the prediction of lipase structure. For instance, Ofiţeru et al. suggested that AlphaFold2 can be used for the identification and classification of millions of protein sequences from metagenomics sequencing and thus obtaining 78 putative cold-adapted bacterial lipase sequences[87]. Moreover, Oberer and co-workers indicated that three-dimensional models of adipose triglyceride lipase demonstrated that the mapping of important amino acids can strongly corroborating the significance of these residues[88].
- 4, Computer-aided methods for thermostability engineering of lipase.
It would be better to add descriptions of ancestral mutation methods, etc. (https://doi.org/10.1016/j.jbc.2022.102435).
Thanks so much for your kindly comments, the descriptions of ancestral mutation methods was added in this version.
Gillam et al. suggested that the ancestral sequence reconstruction can generate highly stable folds and thus improving the thermostability of enzymes, due to the ancestral proteins were frequently more thermostable than their extant descendants[117]. Up to date, the ancestral sequence reconstruction method has been used for improving stability of many enzymes, such as lipase, nucleoside diphosphate kinase, 3-isopropylmalate dehydrogenase et al[118, 119]. The main processes of ancestral sequence reconstruction mainly includes: nucleic acid/amino acid sequence collection of modern enzymes, multiple sequence alignment, phylogenetic tree construction, computer prediction of ancestral enzyme sequences, gene cloning, and enzymatic property characterization[120]. Among them, the ClustalW and Multiple sequence comparison by log-expectation (MUSCLE) were the most widely used multiple sequence alignment programs and the MUSCLE exhibited great advantages in speed and accuracy.
- Please first define "hbonds" somewhere like "hydrogen bonds (hbonds)".
Thanks so much for your kindly comments, the full name of hydrogen bonds (hbonds) has also been added in P14.
- What is "romantic"? aromatic?
Thanks so much for your kindly comments and we are sorry for the mistakes. The "romantic" has been revised to “aromatic”.
Reviewer 2 Report
The current review articles report an overview about the food applications of lipases as well as the computer-aided lipases engineering for improving their stability and activity. The review article is well structured and is written in a simple way that most of readers can understand. However, there are many concerns that should be considered before its consideration as below:
1. Abstract, in the sentence “As one of the most widely used biocatalysts, lipases have exhibited their extreme advantages in esterification, amidation, and transesterification reactions”, there are many other processes that lipase exhibited advantages. I suggest you modify to “……in many processes such as esterification, …….”
2. After going through the whole article, it should be noted that the reason of adding each reference should be clearly known. Therefore, please eliminate multiple references. After that, please check the manuscript thoroughly and eliminate the lumps in the manuscript. This should be done by characterizing each reference individually and by mentioning 1 or 2 phrases per reference to show how it is different from the others and why it deserves mentioning. Multiple references are of no use for a reader and can substitute even a kind of plagiarism, as sometimes authors are using them without proper studies of all references used. In the case, each reference should be justified by it is used and at least short assessment provided. Please avoid lump reference (2 or more references together). Explain them individually. Example for this is section 1, ……….to be environmentally unfriendly [1–4], and ………… plastic degradation [12–16]
3. Section 2: I suggest to cancel the following sentence, in decades, with the improvement of the standard of living, the demand for oil is also getting high and high, not only delicious, and healthy, but also more and more special”.
4. The title of table 2 is (Several studies of engineering of solvents tolerance of lipases), however the authors did not well connected the explanation of the table with its title/content, please revise the sentences start with “As shown in Table 2 ,several kinds of lipases can be used in modification of lipids, especially Novozyme435.Moreover, various kinds of reaction” and rewrite to appropriately reflect the content of the table.
5. In the introduction section, the first sentence start with “With the depletion of nonrenewable resources and the increase in environmental pollution, organic reactions, which produced a lot of organic solvent consumption,…….”, it is advised to highlight the disadvantage of chemical esterification such as using toxic and corrosive catalysts such as sulfuric acid/sodium hydroxide, as well as using an elevated reaction temperatures. Both results in generating random non selective reaction. You can then compare these by the green enzymatic process. Such sequence would grab the reader attention. With reference to the advantages of the enzymatic process, I suggest to cite the following article 10.1016/j.scp.2022.100690
6. From 146 references, only 1 (2023) reference is cited, please cite more 2023 journals.
7. Figure 2: the bibliometric study has been established last June 2022, with overlocking the 2022 statistics (from 2010 to 2021), now it already passed more than a year, I suggest to update your study by including the 2022 articles in this bibliometric study. Please update the number of papers and the related explanation in the manuscript accordingly.
8. Please go thought the whole manuscript and make sure to change all the names of the strains to Italic, even in reference list.
9. This article contains many grammatical errors. Kindly revise and update accordingly.
10. Figures 5, 6, and 7, are they original? the authors need to request copyright where applicable.
This article contains many grammatical errors. Kindly revise and update accordingly.
Author Response
Reviewer 2:
The current review articles report an overview about the food applications of lipases as well as the computer-aided lipases engineering for improving their stability and activity. The review article is well structured and is written in a simple way that most of readers can understand. However, there are many concerns that should be considered before its consideration as below:
- Abstract, in the sentence “As one of the most widely used biocatalysts, lipases have exhibited their extreme advantages in esterification, amidation, and transesterification reactions”, there are many other processes that lipase exhibited advantages. I suggest you modify to “……in many processes such as esterification, …….”
Thanks so much for your kindly comments, it was revised to “As one of the most widely used biocatalysts, lipases have exhibited their extreme advantages in many processes such as esterification, amidation, and transesterification reactions, which made them widely used in the food industrial productions.”
- After going through the whole article, it should be noted that the reason of adding each reference should be clearly known. Therefore, please eliminate multiple references. After that, please check the manuscript thoroughly and eliminate the lumps in the manuscript. This should be done by characterizing each reference individually and by mentioning 1 or 2 phrases per reference to show how it is different from the others and why it deserves mentioning. Multiple references are of no use for a reader and can substitute even a kind of plagiarism, as sometimes authors are using them without proper studies of all references used. In the case, each reference should be justified by it is used and at least short assessment provided. Please avoid lump reference (2 or more references together). Explain them individually. Example for this is section 1, ……….to be environmentally unfriendly [1–4], and ………… plastic degradation [12–16]
Thanks so much for your kindly comments, some references which remarked as 1-4 in this paper is reference that all of these references can support our conclusions, and many other publications are also added their references like this. Moreover, we have deleted some of them in this version to eliminate multiple references as your kindly suggestions.
- Section 2: I suggest to cancel the following sentence, in decades, with the improvement of the standard of living, the demand for oil is also getting high and high, not only delicious, and healthy, but also more and more special”.
Thanks so much for your kindly comments, we have deleted it as your suggestions.
- The title of table 2 is (Several studies of engineering of solvents tolerance of lipases), however the authors did not well connected the explanation of the table with its title/content, please revise the sentences start with “As shown in Table 2, several kinds of lipases can be used in modification of lipids, especially Novozyme435. Moreover, various kinds of reaction” and rewrite to appropriately reflect the content of the table.
Thanks so much for your kindly comments, we have changed it to “As shown in Table 2, several studies of engineering of solvents tolerance of lipases were displayed and suggested that there are several strategies can be used for imrpoving solvent tolerance of lipase.” In this version.
- In the introduction section, the first sentence start with “With the depletion of nonrenewable resources and the increase in environmental pollution, organic reactions, which produced a lot of organic solvent consumption,…….”, it is advised to highlight the disadvantage of chemical esterification such as using toxic and corrosive catalysts such as sulfuric acid/sodium hydroxide, as well as using an elevated reaction temperatures. Both results in generating random non selective reaction. You can then compare these by the green enzymatic process. Such sequence would grab the reader attention. With reference to the advantages of the enzymatic process, I suggest to cite the following article 10.1016/j.scp.2022.100690
Thanks so much for your kindly comments, we have added the sentence “Moreover, the traditional chemical esterification often uses toxic and corrosive catalysts such as sulfuric acid/sodium hydroxide, as well as using an elevated reaction temperature, which will also results in generating random non selective reaction.” in our introduction as your suggestions. Moreover, the 10.1016/j.scp.2022.100690 was cited in this version.
- From 146 references, only 1 (2023) reference is cited, please cite more 2023 journals.
Thanks so much for your kindly comments, we have added several 2023 papers in this version.
For instance, https://doi.org/10.3390/microorganisms11020510 and https://doi.org/10.1016/j.cej.2023.142011 et al.
- Figure 2: the bibliometric study has been established last June 2022, with overlocking the 2022 statistics (from 2010 to 2021), now it already passed more than a year, I suggest to update your study by including the 2022 articles in this bibliometric study. Please update the number of papers and the related explanation in the manuscript accordingly.
Thanks so much for your kindly comments, the data of Figure 2 was updated in this version.
- Please go thought the whole manuscript and make sure to change all the names of the strains to Italic, even in reference list.
Thanks so much for your kindly comments, the data of Figure 2 was updated in this version.
- This article contains many grammatical errors. Kindly revise and update accordingly.
Thanks so much for your kindly comments, the scientific names of all organisms involved in this paper were revised to italics in this version.
- Figures 5, 6, and 7, are they original? the authors need to request copyright where applicable.
Thanks so much for your kindly comments, the copyright information of Figure 5 and Figure 7 were added in this version, and Figure 6 is original figure drawn by the authors.
Round 2
Reviewer 1 Report
This manuscript is well enough revised, but the English should be checked by a native speaker.
Author Response
Thanks so much for your kindly comments, the English has been thoroughly revised in this version.
Reviewer 2 Report
The authors have solved some of the concerns that were raised by the reviewer, however two issues still exist:
1. Again for figure 2: please update the study to include the range from 2010 to 2022), instead of (2010 to 2021).
2. For figures 5 and 7, the copyright issue still unclear, have the authors contacted the publishers to ask about the copyright of the copied figures? The copyright is usually taken place automatically through specific portals at the publisher site/paper link. You should then receive a number for every granted copyright.
Author Response
The authors have solved some of the concerns that were raised by the reviewer, however two issues still exist:
- Again for figure 2: please update the study to include the range from 2010 to 2022), instead of (2010 to 2021).
Thanks so much for your kindly comments, the data of Figure 2 was updated in this version.
- For figures 5 and 7, the copyright issue still unclear, have the authors contacted the publishers to ask about the copyright of the copied figures? The copyright is usually taken place automatically through specific portals at the publisher site/paper link. You should then receive a number for every granted copyright.
Thanks so much for your kindly comments, the Figure 5 a,b and Figure 7c, is open access and donot need permission for reproduction. For Figure c,d and Figure 7a,b,d permissions are as following:
